# Positivity Rate Investigation and Anthelmintic Resistance Analysis of Gastrointestinal Nematodes in Sheep and Cattle in Ordos, China

**DOI:** 10.3390/ani12070891

**Published:** 2022-03-31

**Authors:** Bin Hou, Rong Yong, Jiya Wuen, Yong Zhang, Buhe Buyin, Dihua Subu, Huhen Zha, Hong Li, Surong Hasi

**Affiliations:** 1Key Laboratory of Clinical Diagnosis and Treatment Technology in Animal Diseases, Ministry of Agriculture, College of Veterinary Medicine, Inner Mongolia Agricultural University, Hohhot 010018, China; hb960906@126.com (B.H.); 18698434457@163.com (J.W.); 2Ordos Animal Disease Prevention and Control Center, Ordos 017000, China; yongrong5145@126.com (R.Y.); zhangyong19801224@163.com (Y.Z.); 3Wushen Animal Disease Prevention and Control Center, Ordos 017300, China; baoyintian@126.com; 4Hangjin Animal Disease Prevention and Control Center, Ordos 017400, China; su849725982@163.com; 5Otog Animal Disease Prevention and Control Center, Ordos 016100, China; eqykzxggyx@163.com; 6Ejin Horo Animal Disease Prevention and Control Center, Ordos 017200, China; mengkemengke@126.com

**Keywords:** gastrointestinal nematodes, positivity rates, sheep, cattle, anthelmintic resistance

## Abstract

**Simple Summary:**

In order to understand the positivity rates of gastrointestinal nematodes in cattle and sheep in Ordos, and the effects of different pasture types on the distribution of gastrointestinal nematodes, we conducted an epidemiological investigation and analysis in four banner districts of Ordos. The results showed that the positive rates of sheep and cattle were 38.84% and 4.48%, respectively. The anthelmintic resistance analysis revealed that the nematode population in the area was severely resistant to ivermectin and albendazole, and resistance to levamisole, nitroxynil and closantel was suspected.

**Abstract:**

Gastrointestinal nematodes (GINs), such as *Trichostrongylidae*, are important pathogens in small ruminants, causing significant losses in these livestock species. Despite their veterinary importance, GINs have not been studied in certain regions of the world. Therefore, much of their epidemiology and economic impact on production remain unknown. In the present study, a systematic epidemiological survey based on the modified McMaster technique was conducted to investigate the type and infection of GINs in sheep and cattle. In 9622 fecal samples from 491 sampling sites in the four main banner districts of Ordos, the prevalence of GIN infection was found to be 38.84% and 4.48% in sheep and cattle, respectively. At the same time, the effects of four pasture types on the distribution of GINs were analyzed. This study also found severe resistance to ivermectin and albendazole in GINs and suspected anthelmintic resistance in nitroxynil, levamisole and closantel. We report the type and infection of GINs in Ordos, with the aim to help the prevention and control of GINs. Based on the results of the questionnaire survey and GIN resistance test, we found several reasons for the anthelmintic resistance of GINs, consequently providing new ideas for controlling the occurrence of anthelmintic resistance.

## 1. Introduction

Gastrointestinal nematodes (GINs) are the cause of major losses in all animal production systems worldwide [1,2]. Large amounts of GINs can infect a variety of hosts, ranging from large herbivores to small companion animals [3]. In small ruminant livestock, GINs cause production losses in domestic animals in terms of treating clinical cases, subclinical infections and mortality. The effects are less serious in cattle, but possibly include a variable impact on milk production and growth retardation [4,5]. The often indiscriminate and intensive use of anthelmintics has resulted in the selection of anthelmintic-resistant nematodes, which are increasingly reducing the effectiveness of pharmaceutical-based control [6].

Understanding the distribution and infections of pathogens is the first step for the prevention and control of parasitic infections and diseases; in particular, subsequent investigation and modeling has the most influence [7,8]. While studies involving many parasite species can be found, the majority have focused on relatively few species, principally *Haemonchus contortus* and *Ostertagi*. Consequently, our understanding of the parasite regulation and dynamics of many parasite species remains relatively rudimentary [9]. Currently, anthelmintic resistance has been reported in China [10], the United States [11], New Zealand [12], Mexico [13], Australia [14] and Brazil [15]. However, there is no complete alternative to anthelmintic therapy; it is therefore necessary to utilize chemotherapeutic treatments to control GINs. Macrocyclic lactones have dominated the world market for the past 50 years, but their efficacy has severely declined over time. In some parts of the world, anthelmintic combinations have been introduced to mitigate the occurrence of anthelmintic resistance [16,17]. In Australia, New Zealand and China there are a great number of anthelmintic combination products registered. However, there are no anthelmintic combination products in the United States that consider the presence of refugia to prevent the exacerbation of multi-drug resistance [18].

Considering that domestic animals are the most important economic pillar of Ordos, there are a relatively higher number of pasture types in this area. Thus, the relationship between nematode distribution and the environment can be studied, and the increasing amount of resistance to anthelmintics appearing in clinical treatment can be examined. The present study aims to describe the infection and anthelmintic resistance of GINs in cattle and sheep in Ordos.

## 2. Materials and Methods

### 2.1. Collection of Sample

From July to October 2021, in four banner districts, according to the administrative regions and topographic features of Ordos, four sampling points were set up (Figure 1). A total of 10,126 fecal samples of sheep (*n* = 8542) and cattle (*n* = 1584) were collected from the sampling area; meanwhile, 1239 epidemiological questionnaires were collected. In this study, fecal samples of all the research animals were naturally infected. All the samples were placed into disposable airtight sealed pockets marked with basic information (e.g., the collection area, basic aquaculture information, and history of anthelmintic use), immediately transported to the parasitology laboratory of Ordos Animal Disease Prevention and Control Center under low-temperature conditions, and stored in a refrigerator at 4 ℃ in order to avoid the impact of egg hatching on the judgment of results until identification.

Anthelmintic resistance analysis: Based on the epidemiological survey, to evaluate the anthelmintic resistance (albendazole tablets, ivermectin injection, levamisole tablets, nitroxynil injection and closantell injection), four groups of naturally infected sheep were selected as the fixed-point study sample. A total of 480 severely infected sheep (EPG > 1500) were selected and then divided into 6 serpentine groups according to EPG size, with 80 sheep in each group. All the anthelmintics had passed the content determination of the *Veterinary Pharmacopoeia of the People’s Republic of China.* Fecal samples were collected from each household before and 14 days after treatment and analyzed using a modified McMaster technique.

### 2.2. Identification of Nematode Eggs

The species identification of GINs was based on the morphological characteristics and typical structure of different eggs, with reference to the pictures in the *Color Atlas of Morphological Classification of Livestock and Poultry Nematodes in China*. This was also confirmed by the structure of the larvae cultured in feces.

### 2.3. Fecal Eggs Quantitative Examination

Fecal egg counts were undertaken using the modified McMaster technique, as described in *Veterinary Clinical Parasitology*. The reduction in fecal egg counts (FECR) was calculated according to the following formula:FECR(%) = (EPG_pre-treatment_ − EPG_post-treatment_)/EPG_pre-treatment_ × 100%(1)

### 2.4. Statistical Analysis

Differences in the infection of nematodes among geographical areas and sampling sites were analyzed by using the one-way ANOVA test within the software SPSS V 25 (IBM, New York, NY, USA), and the differences were considered significant when *p* value < 0.05.

The results of the anthelmintic resistance test were calculated by the following formulae according to the changes in EPG pre-treatment and post-treatment.
S_i_^2^ = [∑_j_X_ij_^2^ − (∑_j_X_ij_)^2^/n_i_]/(n_i_ − 1)(2)
(3)$$Y^{2}={S_{t}}^{2}/(n_{t}{\bar{x}}_{t^{2}})+{S_{c}}^{2}/(n_{c}{\bar{x}}_{c^{2}})$$
(4)$$\mathrm{Upper}\;\mathrm{CL}:100\;\{1-({\bar{x}}_{t}/{\bar{x}}_{c})\;\exp\;(-2.048\;\sqrt{Y^{2}})]\}$$
(5)$$\mathrm{Lower}\;\mathrm{CL}:100\;\{1-({\bar{x}}_{t}/{\bar{x}}_{c})\;\exp\;(+2.048\;\sqrt{Y^{2}})]\}$$
where i denotes either the treated (t) or control (c) groups, j denotes each sheep in the group, $\bar{x}$ denotes the post-treatment EPG arithmetic mean, n denotes either number of test animals, S_i_^2^ denotes the variance of the arithmetic scale and *Y*^2^ (log scale) denotes the variance of the mean egg number reduction in post-treatment. Resistance was present if: (i) the percentage reduction in egg count was less than 95%; and (ii) the 95% confidence level was less than 90%. If only one of the two criteria were met, resistance was suspected [19]. 

## 3. Results

In the present study, the overall positive rates of GIN infection in sheep and cattle were 38.84% (3318/8542) and 4.48% (71/1584), respectively. As shown in Table 1, the Wushen Banner, Ejin Horo Banner, Otog Banner and Hangjin Banner showed sheep with GIN positivity rates of 32.16% (898/2792), 6.71% (38/566), 49.28% (749/1520) and 2% (1/50), respectively. The cattle with GIN positivity rates were 33.48% (770/1930), 5.13% (24/468), 46.68% (901/1930) and 2% (8/400), respectively. The differences between the four regions were not statistically significant (*p* > 0.05).

The identification of various GINs is shown in Figure 2. The principal species infecting sheep were *H.contortus* and *Nematodirus*, and the positivity rates were 58% and 15.76%, respectively (Figure 3).

The principal species infecting sheep were *H.contortus* and *Chabertia* spp., and the positivity rates were 3.16% and 4.61%, respectively.

The geography and various topographies of Ordos, including sandy pastures, hilly pastures, low-lying pastures and pen-fed pastures, were considered to determine whether there were differences between pasture types and GIN infections. As shown in Figure 4, the GIN infection rate was similar among sandy pastures, hilly pastures and low-lying pastures, but the infection rate of pen-fed pastures was low.

As shown in Figure 5, 59.39% of herdsmen used macrocyclic lactone for deworming, and 11.89% used anthelmintic combinations.

Based on previous epidemiological investigation, we screened 480 severely infected sheep (EPG > 1500) for anthelmintic resistance testing. The results in Table 2 show that the widely used albendazole and ivermectin have developed anthelmintic resistance, and resistance to levamisole, nitroxynil and closantel was suspected. Although the medication history use of nitroxynil and closantel is only three years, a trend of anthelmintic resistance was still found.

## 4. Discussion

GINs infection are a common constraint in domestic animals that can cause a decrease in animal health, productivity and farm profitability [20]. Although some progress has been made in parasite vaccination, efforts are still needed. Since there is no alternative method comparable to anthelmintics, it is still necessary to utilize chemotherapeutic treatments to control GINs.

In the present study, the infection rates of GINs in sheep in the Wushen Banner and Otog Banner were 32.16% (898/2792) and 33.48% (770/1930), respectively, which are lower than Ejin Horo Banner, 49.28% (749/1520), and the Hangjin Banner, 46.68% (901/1930). Because herdsmen in Ordos often report that their deworming drugs are ineffective, Professor Hasisurong started focusing on local parasitic studies in 2017, discovered the anthelmintics resistance of local GINs in the Wushen Banner, and screened for appropriate anthelmintics. Compared with the initial data from 2017, the infection rate of GINs has decreased from 84.3% to 32.16% [21]. However, the Hangjin Banner and Ejin Horo Banner lack relevant professional training, and herders have little awareness of parasite control. However, the infection rate of GINs in sheep in Ordos is lower than that in Ethiopia (83%) [22], Bangladesh (77.1%) [23], South Africa (81%) [24] and Algeria (96%) [25]. Although the different breeds, detection methods, geographical differences and sample sizes are factors that may contribute to varying infection [26], the infection rate of GINs in Ordos is still low. The GIN infection rate of cattle was found to be lower than that in sheep; different GINs likely have different susceptibility to different hosts [27]. Some GINs can produce acquired immunity and be maintained for a long time after infection in cattle [28], and cattle are usually kept separately from sheep. In addition, soil type may have a major effect on the ability of larvae to migrate. The predilection of larvae to remain relatively close to the fecal pat may have a substantial impact on transmission, as cattle do not graze close to fecal pats until foraging is very limited. Regarding different pasture types, we can see that the infection levels of sandy pastures, hilly pastures and low-lying pastures were similar, and there was no significant difference, but the infection intensity of nematodes in the pen-fed pastures was very low. The level of pen-fed management was high, and the prevention and control of parasites were also effectively performed, which is also consistent with the results reported by the frontal Eye L in 2018 [21]. There are also reports that, in captive areas, dung-burying beetles, coprophagous beetles and earthworms can greatly reduce the larvae of some trichostrongylids in pastures. They contribute to the spread of the fecal material in the pasture and cause larval death as a consequence of drying [29].

At the beginning of the 1970s, resistance to the first BZ anthelmintic, thiabendazole, was reported [30]. In the following decade, resistance to pabendazole, fenbendazole, oxfendazole and ivermectin was reported. In the present study, severe anthelmintic resistance was observed to albendazole and ivermectin. Resistance to levamisole, nitroxynil and closantel was suspected. The anthelmintic resistance of macrocyclic lactone and benzimidazole was very common. Although the fecal egg reduction rate of benzimidazole and ivermectin was approximately 50%, they still lost the characteristics of high efficiency, safety and broad spectrum. According to the review, in Ordos, albendazole and ivermectin were introduced in 1984 and 1988, respectively. More than 30 years of continuous use has caused serious anthelmintic resistance problems. To date, 70.35% of herders still use these anthelmintic. However, nitroxynil and closantel have been used locally for only three years, but resistance is already starting to be suspected and warrants attention.

Based on the questionnaire, we found that 60.23% (727/1202) of herders would choose anthelmintics based on veterinary store recommendations (Figure 6), while 41.44% (489/1180) were dissatisfied with the effectiveness of anthelmintics (Figure 7), contrary to the results of studies in the UK, Belgium and Ireland [31,32], as the problem of anthelmintic resistance in Ordos is more serious. In fact, 85.31% (941/1103) of herders will receive treatment twice a year (Figure 8). However, parasitological analysis was not used prior to treatment and was evaluated later in the treatment schedule, which seems to be a common feature among herders in Ordos. This is similar to the results of the 1.5 insect repeats performed annually in Norwegian herders every year; however, Norwegian herders performed more prophylactic insect repeats, and 53% did not experience gastrointestinal nematode problems [33]. The survey also found that 46.38% (508/1163) of herders would increase the anthelmintics dose based on recommendations or without authorization, fundamentally contributing to the development of anthelmintic resistance(Figure 9).

Through this experiment, we identified many problems. One problem is that herders are deworming year after year. Many grassroots veterinary pharmacy staff do not have a solid enough professional foundation to give effective advice. During the deworming process, we performed more blind prophylaxis, which put some pharmacological pressure on some groups. In terms of operation, medication is administered at the group level, not the individual level. Domke’s previous research showed that most shepherds (79%) estimated the appropriate amount of insect repellent according to the visual evaluation of sheep weight, but visual evaluation is not accurate [34]. The number of antiparasitic agents available to veterinary practitioners is relatively limited. However, the problem of anthelmintic resistance is worsening year by year. In the author’s opinion, to solve the problem of anthelmintic resistance, it is necessary not only to simply invent new technologies and methods, but also to improve the quality of industry personnel and strengthen the management of future breeding techniques.

## 5. Conclusions

The present study reported the infection of GINs in sheep and cattle in Ordos. The infection rates of sheep and cattle were 38.84% and 4.48%, respectively. The infection rates also varied among different pasture types, with the lowest infection rate in the pen-fed pastures. Meanwhile, five commonly used anthelmintics containing ivermectin and albendazole had severe resistance, and anthelmintic resistance to levamisole, nitroxynil and closantel was also suspected. Through data analysis and a questionnaire survey, we found several reasons for anthelmintic resistance in GINs. This can provide new ideas for controlling the occurrence of anthelmintic resistance.

## Figures and Tables

**Figure 1 animals-12-00891-f001:**
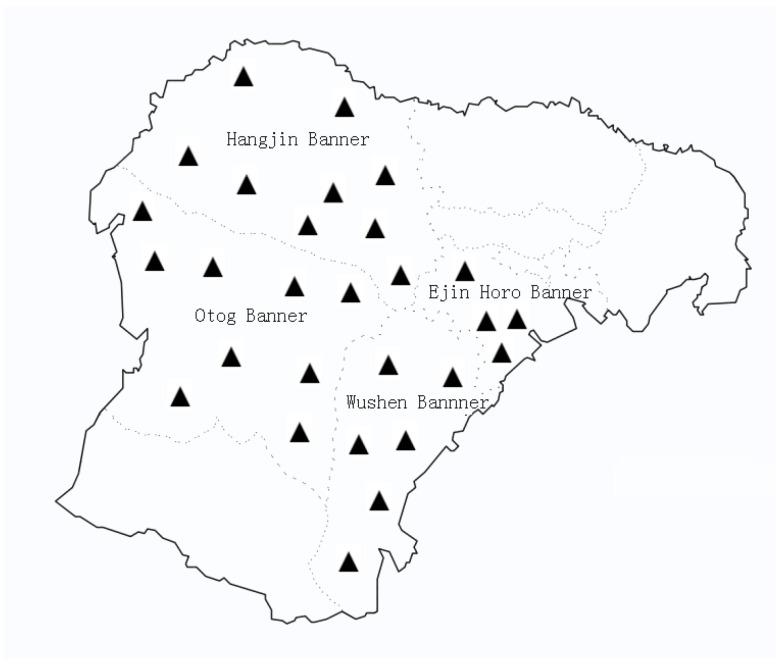
Geographical distribution of sampling sites in the present study (▲).

**Figure 2 animals-12-00891-f002:**
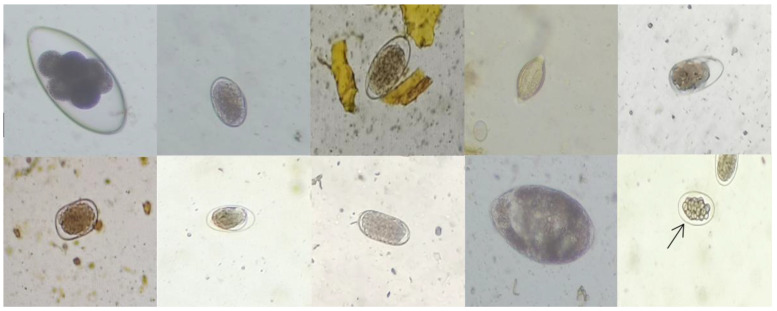
GIN eggs are, from left to right, *Nematodirus* egg, *Haemonchus contortus* egg, *Chabertia* spp. egg, *Trichuris* spp. egg, *Trichostrongylus* spp. egg, *Bunostomum* spp. egg, *Oesophagostomum* spp. egg, *Cooperia* spp. egg, *Marshallagia* spp. egg and Osertagia spp. egg (10 × 40).

**Figure 3 animals-12-00891-f003:**
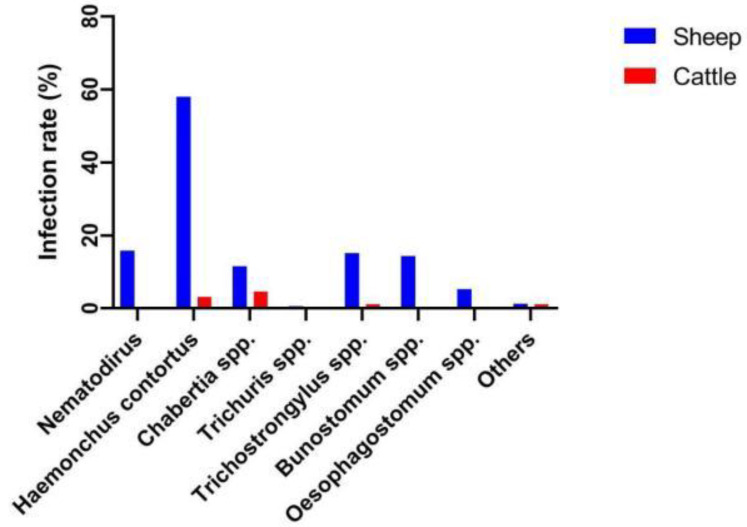
GIN species and infection rates in sheep and cattle.

**Figure 4 animals-12-00891-f004:**
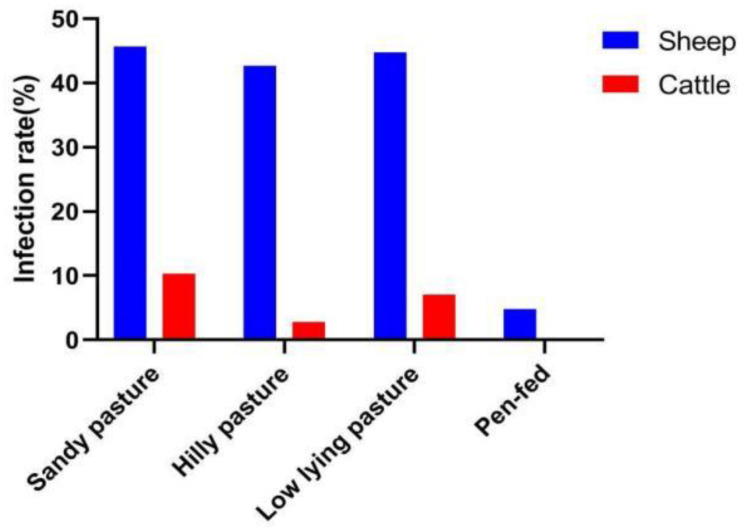
Infection of sheep and cattle with nematode disease in different grassland types.

**Figure 5 animals-12-00891-f005:**
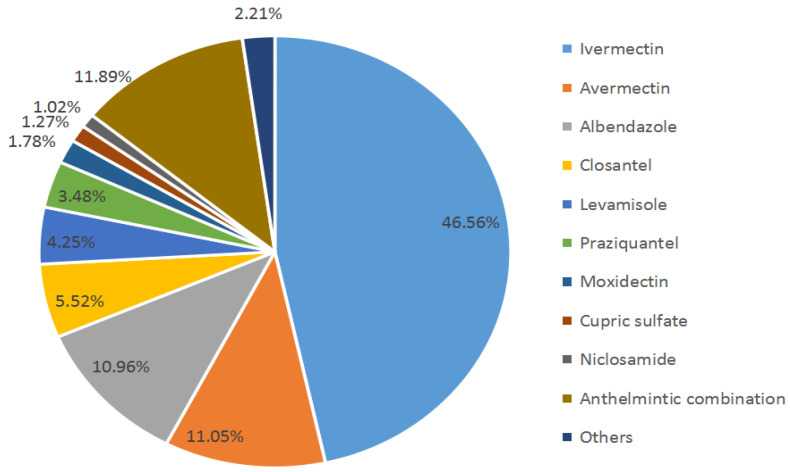
Herdsmen’s choice of anthelmintic.

**Figure 6 animals-12-00891-f006:**
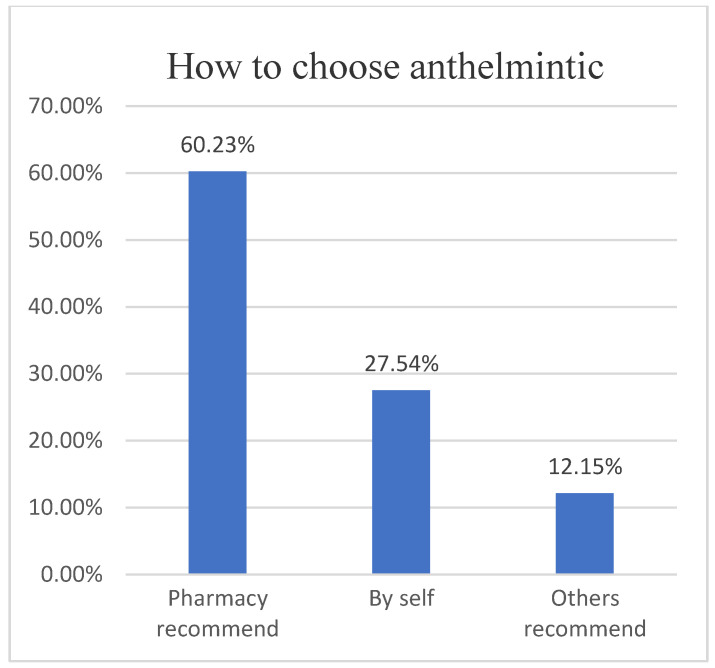
How do herdsmen to choose anthelmintic.

**Figure 7 animals-12-00891-f007:**
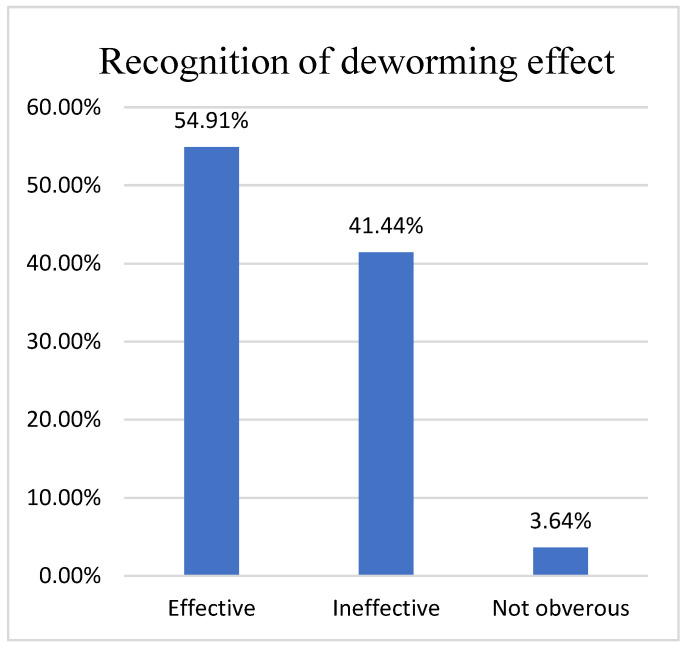
Herdsmen’s recognition of deworming effect.

**Figure 8 animals-12-00891-f008:**
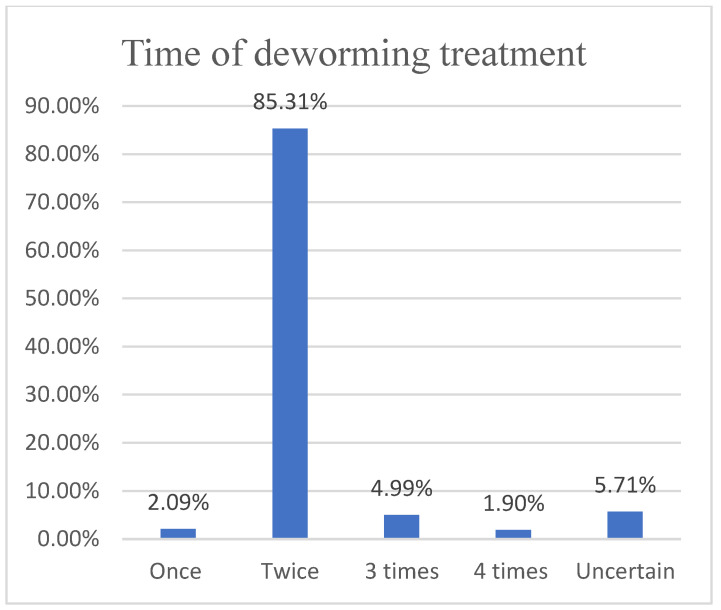
Herdsmen’s treatment time for deworming.

**Figure 9 animals-12-00891-f009:**
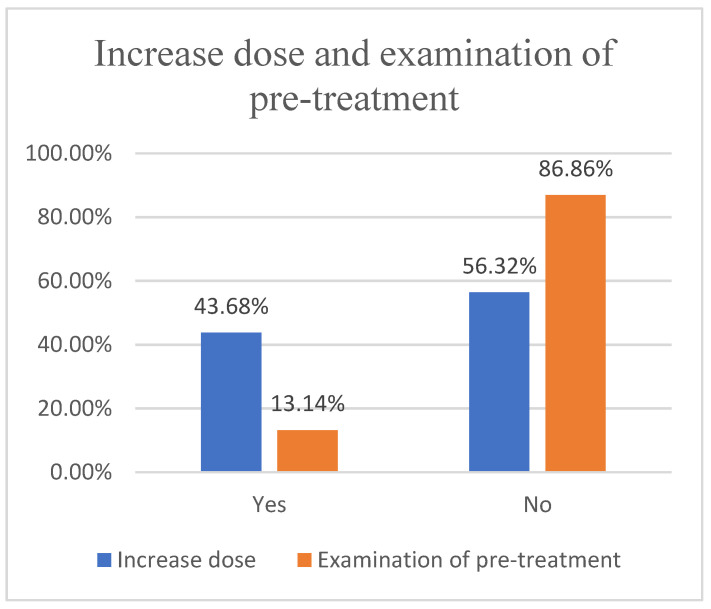
Whether it will increase dose and examination of pre-treatment.

**Table 1 animals-12-00891-t001:** Infection of GINs of sheep and cattle in different areas of Ordos.

Geographical Areas	No. of Epidemiological Questionnaires	No. of Sheep	No. of Cattle	No. of Positive Sheep/Cattle (%)
Wushen Banner	161	2792	566	32.16/6.71
Ejin Horo Banner	300	1520	50	49.28/2
Otog Banner	300	2300	468	33.48/5.13
Hangjin Banner	177	1930	400	46.68/2
Total	938	8542	1584	38.84/4.48

**Table 2 animals-12-00891-t002:** Results of resistance tests of anthelmintics.

Anthelmintic	*n*	Pre-Treatment EPG (Mean ± sd)	14th Day Post-Treatment EPG (Mean ± sd)	FECR (%)	95% Upper CL	95% Lower CL	Judge
Albendazole	80	1971 ± 1379	545 ± 719	71.2	76.9	53.14	Resistance
Ivermectin	80	1985 ± 1635	916 ± 1192	47.23	59.4	17.64	Resistance
Levamisole	80	2163 ± 1741	87 ± 344	93.33	96.08	92.05	Suspected
Nitroxynil	80	2069 ± 1400	64 ± 225	94.89	98.32	90.56	Suspected
Closantel	80	1942 ± 1579	49 ± 229	93.87	99.01	90.82	Suspected
Control	80	2158 ± 1472	1695 ± 1224	17.42	-	-	-

Citations: EPG (Egg Per Gram), FECR (Fecal Egg Reduction Rate), CL (Confidence Level).

## Data Availability

Data are contained within the article.

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
