# Peer review of "Positivity Rate Investigation and Anthelmintic Resistance Analysis of Gastrointestinal Nematodes in Sheep and Cattle in Ordos, China"

_animals, 2022, doi:10.3390/ani12070891_

Round 1

Reviewer 1 Report

This manuscript describes the prevalence and phenotypic of the main gThis manuscript describes the prevalence and phenotypic of the main gastrointestinal nematodes of a China region. The information presented in this paper is relevant in the field of veterinary parasitology in the order to identify the main parasites in domestic animals to establish adequate strategies for GIN control. However, I consider some limits in this study, I think that the authors might perform a genotypic analysis of anthelmintic resistance in both sheep and cattle in the anthelmintics benzimidazole and ivermectin. On the other hand, is important make mention that the nematode H. contortus is exclusive to small ruminants. Although, there are reports where the cattle have been artificially infected with H. contortus, in natural conditions it is difficult. 

Some comments and suggestions

Simple summary

L21: Does this positive rate correspond to the prevalence of GIN in both sheep and cattle?. Please, clarify it

Material and methods

L101: The authors should describe more details on the identification of the infective larvae. What methodology was used for the phenotypic identification of the larvae?
Why was not used molecular techniques like PCR, to corroborate the GIN genera? 

Results

Figure 2: The identification of Haemonchus, Chabertia, Cooperia, and Ostertagia through eggs is very complex due to the similarity among genera.  If the authors have images of the infective larva of these nematodes eggs should be replaced by these images. Now, do these GIN eggs correspond to the fecal samples of sheep?. Please, clarify it.

Figure 3: Is important to clarify that the Haemonchus contortus is a nematode exclusive of the small ruminants not in Cattle. Thus, Figure 3 can be modified considering my comment.

Reviewer 2 Report

All the manuscript requires more clarity, and English needs edition: for instance, Lines (43-45)that…that; ... with by…so on

Line 3. It should be written in plural

Line 25. A family of nematodes should not be written with spp.

Line 30. The infection sounds redundant to prevalence

Lines 54-55. Previous abbreviation, please write full name

Line 60. Macrolides are a class of antibiotics. Perhaps you refer to macrocyclic lactones

Line 67. Odros, a China region?

Line 77. A total of 10,126 animals were included in this study. It's important to mention why you collect less faecal samples. Where they from individual hosts? Or, where they a pool? Or perhaps, no samples were collected. Just mentioned it. Also, authors should define if faecal samples were collected from same place

Line 83. It's important to specify for how long were the eggs stored, just to check their viability. Although, some eggs can be identified without problem, but eggs from Trichostrongylidae family is not easy.

Line 98. The identification of species of GIN could be misleading, based on a morphological analysis using eggs. Use the identification of infective larvae is the most accurate taxonomic from morphological characters in these stages.

Lines 130- 138. Results require the climate dominant per region, and data should be presented in table only, into the text was repetitive this information

Line 141. Authors mentioned HELMINTHS and the gastrointestinal nematodes Haemonchus and Nematodirus were written as results. Review your information. Moniezia is cestode, that is a helminth. Sorry, did you identify Cooperia in cattle? This is a important nematode in temperate and tropical climate, so, for this reason it is important to mention which climate was in each region

Line 198. This statement can be misleading because cattle infection indicates in this study was H. contortus. This is possible, and you can check information from Brasilian Researchers who identified H. contortus and H. place in both ruminant species, sheep and cattle, check this information in NCBI

Line 2016. Review information about macrolides and macrocyclic lactone, please

DISCUSSION. Please, try to improve the discussion related  with your information

Reviewer 3 Report

Authors aims to describe the prevalence and anthelmintic resistance of GIN in cattle and sheep in Odros.

Specific comments/suggestion are highlighted in the attached .pdf file.

General comments:
- The introduction lack of specific data related to GIN in China or positivity rates reported in other countries.
- Kruskal-Wallis or other should be performed to compare positivity rates, or at least check for normal distribution of positivity rates.
- Formal aspects correction of the manuscript should be carried out.

It is important to tackle the lack of knowledge about GIN an other parasite infections in small ruminants and this works helps to solve this issue and the resistance to terapeutics.

Reviewer 4 Report

The manuscript entitled " Epidemiological Investigation and Anthelmintic Resistance Analysis of Gastrointestinal Nematode in Sheep and Cattle in Ordos, China " reported the infection of GIN of sheep and cattle in Ordos. The work analysed GIN infection rates of sheep and cattle, focusing on the emergence of drug resistance, which causes major problems today. The paper is generally well written and structured. Sufficient information about the previous studies findings is presented for readers to follow the present study rationale and procedures. 

The article does an epidemiological survey of the Gastrointestinal nematodes (GIN) analysing resistance to ivermectin and albendazole, and levamisole, nitroxynil and closantel.  The article is well written and easy to read, the information presented in this paper is interesting in the field of veterinary parasitology, even if the paper provides only descriptive results.

Round 2

Reviewer 1 Report

The authors attended to all comments and suggestions. I recommend this manuscript to publish in Animals